# Kinetic Analysis of Thermal Degradation of Recycled Polypropylene and Polystyrene Mixtures Using Regenerated Catalyst from Fluidized Catalytic Cracking Process (FCC)

**DOI:** 10.3390/polym15092035

**Published:** 2023-04-25

**Authors:** Paul Palmay, Leslie Pillajo, Mónica Andrade, Carlos Medina, Diego Barzallo

**Affiliations:** 1Facultad de Ciencias, Escuela Superior Politécnica de Chimborazo ESPOCH, Panamericana Sur Km 1 1/2, Riobamba 060155, Ecuador; 2Facultad Ciencias e Ingeniería, Universidad Estatal de Milagro, Milagro 091050, Ecuador; 3Environmental Analytical Chemistry Group, University of Balearic Islands, Cra. Valldemossa 7.5 Km, 07122 Palma de Mallorca, Spain

**Keywords:** plastic waster, regenerated FCC catalyst, kinetic analysis, thermocatalytic degradation, isoconversional models

## Abstract

The pyrolysis process is a thermochemical recycling process that in recent years has gained importance due to its application in plastic waste, which is one of the biggest environmental problems today. Thus, it is essential to carry out kinetic and thermodynamic analyses to understand the thermocatalytic degradation processes involved in plastic waste mixtures. In this sense, the main objective of this study is to analyze the degradation kinetics of the specific mixture of polypropylene (25%) and polystyrene (75%) with 10% mass of regenerated FCC catalyst which was recovered from conventional refining processes using 3 heating rates at 5, 10 and 15 K min^−1^ by thermogravimetric analysis (TGA). The obtained TGA data were compared with the isoconversional models used in this work that include Friedman (FR), Kissinger Akahira Sunose (KAS), Flynn–Wall–Ozawa (FWO), Starink (ST) and Miura–Maki (MM) in order to determine the one that best fits the experimental data and to analyze the activation energy and the pre-exponential factor; the model is optimized by means of the difference of minimum squares. Activation energy values between 148 and 308 kJ/mol were obtained where the catalytic action has been notorious, decreasing the activation energy values with respect to thermal processes.

## 1. Introduction

Currently, the solid waste generated at the urban and industrial levels has intensified in recent years, plastics being one of the wastes of greatest concern worldwide due to several factors that include the high consumption of resources for their production, their high daily use and their long degradation time due to their polymeric structure [1]. Borelle et al. [2] indicate that by the year 2050, there will be a generation of about 25,000 million metric tons of plastic waste, of which 36.4% will end up in landfills or in the environment; a similar fraction will be incinerated, and only 27.2% will be recycled. Thus, it is essential to search for new sustainable technological alternatives such as pyrolysis that contribute to environmentally adequate management of plastic waste [3,4,5]. Pyrolysis is a thermochemical process that involves the degradation of plastics at high temperatures in the absence of oxygen to obtain valuable products that include liquid and gaseous hydrocarbons, which can occur with or without the presence of a catalyst [6]. The main advantage of catalytic pyrolysis is accelerating the reaction rate by reducing the activation energy of the process, which favors this process by increasing its efficiency [7]. In addition, this process can be carried out on both individual plastics and plastic mixtures; however, its kinetic behavior may vary depending on their composition, which causes additional complexity due to the possible interaction between the different types of plastics that make up the mixture and the formation of additional degradation products [8]. In this way, Aboulkas et al. [9] determined the activation energy for different plastic residues considered the most abundant in domestic garbage, which include high-density polyethylene (HDPE), reporting values between 238 and 247 kJ mol^−1^, polyethylene low density (LDPE) from 215 to 221 kJ mol^−1^ and polypropylene (PP), between 179 and 188 kJ mol^−1^ [6,10,11]. Moreover, polystyrene (PS) is one of the most difficult plastics to recycle due to its lower thermal stability with an activation energy value of around 140 kJ mol^−1^ [10,11]. However, studies of the behavior of plastic mixtures are more limited, PE and PP mixtures being the most studied at different rates and different compositions, where minimum amounts of PP generate a decrease in the activation energy and in the maximum degradation temperature both for HDPE and LDPE, which is due to the synergy effect of the mixture influenced mainly by the lower thermal stability of PP in these composites [12,13]. In addition, Dubdub et al. [14] show the behavior of different plastics and their mixtures where the data obtained are adjusted to a single stage of thermal degradation, and the pyrolysis of pure plastic waste occurs in the order PS < PP < LDPE < HDP, while specific mixtures of PP and PS (50/50, wt%) present lower degradation with activation energy values between 144 and 188 kJ mol^−1^ [15]. In any case, all these studies have been carried out by means of thermal pyrolysis, which shows that there are even more limited studies of the kinetic behavior using catalysts, being a field still to be explored due to its great advantages as mentioned above.

In this sense, to understand the kinetic behavior of plastic waste mixtures when degraded is very important, which is commonly evaluated by thermogravimetric analysis (TGA) under non-isothermal (dynamic) conditions, using different heating rates to determine the general or macroscopic kinetics of the thermocatalytic process [16]. According to the International Confederation of Thermal Analysis Calorimetry (ICTAC) [10,17], the kinetic triplet of the pyrolytic process, i.e., activation energy, reaction order and pre-exponential factor, can be calculated from recommended isoconversional models with the data obtained from the TGA. Different isoconversion models that included Friedman, Kissinger–Akahira–Sunose (KAS) and Flynn–Wall–Ozawa (FWO) have been used to evaluate the kinetic behavior of plastic waste [18] and the method of Criado for the calculation of the reaction mechanism [17]. Thus, the study of kinetics plays a key role in the pyrolysis process on an industrial scale, even when catalytic conditions are handled, where different techniques and catalysts can be used, including those recovered from conventional processes; thus, improving the understanding of the thermocatalytic degradation processes and to develop effective recycling and plastic waste management strategies is essential.

In this context, the main objective of this study is to analyze the kinetic degradation behavior under catalytic pyrolysis conditions from a mixture of recycled PP and PS (25:75, wt%) using a regenerated fluidized catalytic cracking (FCC) catalyst from a petrochemical industry by thermo-gravimetric analysis (TGA). Moreover, the thermodynamic parameters were calculated from the kinetic analysis using 5 isoconversional models at 3 heating rates (5, 10 and 15 K min^−1^) commonly used at the industrial level, whose results were also compared without the use of catalyst. Thus, the design parameters, implementation and operating conditions of catalytic pyrolysis can be evaluated by analyzing its chemical kinetics whose accuracy and precision depend on the reliability in the calculation of its activation energy and pre-exponential factor [9]. Further, the present work represents the first kinetic study in report information on the behavior of the degradation of plastic waste mixtures under the catalytic pyrolysis conditions used to evaluate the relationship between the experimental values of TGA analysis and the corresponding data obtained from the five isoconversion models studied.

## 2. Materials and Methods

### 2.1. Sample Collection

Single-use plastic waste that included polypropylene (PP) and polystyrene (PS) was recycled through completely random sampling of a garbage dump located in Riobamba, Ecuador. These samples were classified, and the PP was separated from the PS. Then, these were washed and crushed to a particle size of 1 mm. The catalyst was collected from a fluidized catalytic cracking (FCC) unit and subsequently regenerated using 200 mL ethanol under stirring at 300 rpm for 14 h, with a ratio of 10 mL of solvent per gram of spent catalyst. Subsequently, the solution was filtered and dried to undergo heat treatment at 350 °C for 1 h with a heating rate of 50 °C h^−1^, then at 450 °C for an additional 1.5 h at 25 °C min^−1^ and finally at 700 °C for 2 h. After that, samples were gradually cooled for further experiments in catalytic pyrolysis process.

### 2.2. Characterization of Plastic Waste

Individual plastic residues were characterized by Fourier Transform Infrared Spectroscopy (FTIR) to determine the composition of functional groups of the sample using a JASCO FT/IR-4100 spectrometer (Quito, Ecuador)The Spectra Analysis program performs the acquisition and treatment of data and provides a numerical value based on the height or area of the peak in a working scan range of 4000 to 580 cm^−1^.

### 2.3. Thermogravimetric Analysis

The kinetic study from recycled plastic waste has been carried out by thermogravimetric analysis (TGA) to obtain mass loss data of the mixed plastic waste with respect to time and temperature. It was performed following the procedure described by Briceno et al. [12] with some modifications. Thus, 25 mg sample (25% PP and 75% PS) with the addition of 10% regenerated catalyst was fed in the METTLER TOLEDO brand TGA-1 (Quito, Ecuador) thermobalance with a precision of ±0.1 mg using three heating rates (5, 10 and 15 K min^−1^) with nitrogen injection at a constant flow of 20 mL min^−1^, heating from room temperature to 1023 K. For data validation, the temperature profiles of each test were performed for triplicate, and these results were corroborated from the first derivate DTG curves.

### 2.4. Kinetic Theory and Model Development

Isoconversional methods are utilized to analyze the degradation kinetics of either pure plastics or plastic waste mixtures. This approach allows for the calculation of the kinetic triplet, assuming that the conversion rates are proportional to the concentration of reactive materials. Thus, the decomposition kinetics can be expressed by the following equation, which takes into account the dependence of the conversion rate on the heating process
(1)$$\frac{d\alpha }{dt}=\beta \frac{d\alpha }{dT}=k\left(T\right)f\left(\alpha \right)$$
where the heating rate is represented by $\beta \left({}^{\circ}C\;{min}^{-1}\right)$; the conversion is α; *f*(*x*) is the function of the kinetics with respect to the conversion; and *k*(*T*) is the kinetic constant function of temperature. The conversion is equal to
(2)$$\alpha =\frac{m_{i}-m_{}}{m_{i}-m_{f}}$$
where $m_{i}$ is the initial mass; $m_{}$ is the mass at a given degradation time; and $m_{f}$ is the final mass. Applying the Arrhenius equation is
(3)$$\beta \frac{d\alpha }{dT}=A\;e^{\left(-\frac{E}{RT}\right)}f(\alpha )$$
where ***E*** represents the activation energy $\left(kJ\;{mol}^{-1}\right)$; ***A*** is the pre-exponential factor $\left(s^{-1}\right)$; and ***R*** is the gas constant $(kJ\;{mol}^{-1}K^{-1}$); the equation shows the relationship between the conversion rate as a function of temperature depending on the rate of heating, which is the starting point for the isoconversional kinetic models that are detailed below.

#### 2.4.1. Method 1: Friedman (FR)

The Friedman method is a differential isoconversion technique, considered the most general since it involves taking natural logarithms on each side of Equation (3), whose result is
(4)$$\ln\left(\frac{d\alpha }{dt}\right)=\ln\left(\beta \frac{d\alpha }{dT}\right)=\ln\left(A\right)-\frac{E}{RT}+\ln\left(f\left(\alpha \right)\right)$$

Applying the contraction cylinder model R2, it is assumed that $f\left(\alpha \right)=2{\left(1-\alpha \right)}^{\frac{1}{2}}$, so Equation (4) can be rewritten as follows:(5)$$\ln\left(\frac{d\alpha }{dt}\right)=\ln\left(\beta \frac{d\alpha }{dT}\right)=\ln\left(A\right)-\frac{E}{RT}+\ln\left(2{\left(1-\alpha \right)}^{\frac{1}{2}}\right)$$

Based on the general formula of the Friedman method in Equation (5), when the expression on the left hand side $\ln\left(\frac{d\alpha }{dt}\right)$ is plotted vs. **1**/***T*** at the same conversion levels, it must be a straight line whose slope and intercept can be used to specify the values of activation energy (***E_a_***) and pre-exponential factor (***A***).

#### 2.4.2. Method 2: Kissinger–Akahira–Sunose (KAS)

The KAS method is an integral isoconversional method based on the Coats–Redfern approximation (b) which uses a fit of Equation (3). This standard equation can be represented as follows:(6)$$P\left(\frac{E}{RT}\right)=\frac{e^{\left(-\frac{E}{RT}\right)}}{{\left(\frac{E}{RT_{m}}\right)}^{2}}=\frac{R^{2}T^{2}e^{\left(-\frac{E}{RT}\right)}}{E^{2}}$$
(7)$$\ln\frac{\left(1-{\left(1-\alpha \right)}^{\frac{1}{2}}\right)}{{T_{m}}^{2}}=ln\frac{AR}{E}-ln\beta -\frac{E}{RT}$$

The activation energy (***E***) and the pre-exponential factor (***A***) are obtained using a linear regression of $\ln\frac{\left((1-{\left(1-\alpha \right)}^{\frac{1}{2}})\right)}{{T_{m}}^{2}}$ vs. **1**/***T***.

#### 2.4.3. Method 3: Flynn–Wall–Ozawa (FWO)

The Flynn–Wall–Ozawa (FWO) method is an integral isoconversional technique that uses the Doyle (c) approximation for **ln*P***(***E***/***RT***).
(8)$$lnP\left(\frac{E}{RT}\right)=-5.331-1.052\frac{E}{RT}$$
(9)$$\ln\left(1-{\left(1-\alpha \right)}^{\frac{1}{2}}\right)=ln\frac{AR}{E}-ln\beta -5.331-1.052\frac{E}{RT}$$

Using Equation (7), the activation energy (***E***) is estimated as the pre-exponential factor (***A***), employing a linear regression of the graph $\ln\left(1-{\left(1-\alpha \right)}^{\frac{1}{2}}\right)$ vs. **1**/***T***.

#### 2.4.4. Method 4: Starink (ST)

Starink performed further analysis and found that if ***A*** is not required to be equal to 1, ***a*** very accurate approximation is given by
(10)$$P\left(\frac{E}{RT}\right)≅\frac{e^{(-1.008\left(\frac{E}{RT}\right)-0.312}}{{\left(\frac{E}{RT}\right)}^{1.92}}$$

Relating the kinetic model with Equations (3) and (10), we have
(11)$$\ln\left(1-{\left(1-\alpha \right)}^{\frac{1}{2}}\right)=ln\frac{AR}{E}-ln\beta -\ln{\left(\frac{E}{RT}\right)}^{1.92}-1.008\left(\frac{E}{RT}\right)-0.312$$

The activation energy (***E***) and the pre-exponential factor (***A***) are estimated using a linear regression for the graph $\ln\left(1-{\left(1-\alpha \right)}^{\frac{1}{2}}\right)$ vs. ***a* 1**/***T***.

#### 2.4.5. Method 5: Miura–Maki (MM)

In the Miura–Maki integral method, the approximation of the temperature integral is the same as that used in the KAS method, Equation (12). To derive the Miura–Maki integral method, it is assumed that the activation energy model is distributable to approximate the right-hand side of Equation (13).
(12)$$1-\alpha ≅e^{\left[\frac{ART^{2}}{\beta E}\;*\;e^{\left(\frac{-E}{RT}\right)}\right]}=\Phi \left(E_{s},T\right)$$

The isoconversional method is under different conditions of temperature increase, so it is assumed that the activation energy corresponding to the same conversion rate remains the same, and the pyrolysis uses a R2 reaction model, obtaining
(13)$$\ln\frac{\left(1-{\left(1-\alpha \right)}^{\frac{1}{2}}\right)}{T^{2}}=ln\frac{AR}{E}-\ln\beta -\frac{E}{RT}$$

The activation energy (***E***) and pre-exponential factor (***A***) are estimated by linear regression of $\ln\frac{\left(1-{\left(1-\alpha \right)}^{\frac{1}{2}}\right)}{{T_{m}}^{2}}vs.$
***a* 1**/***T***.

Once the data treatment was carried out with the equations of each model described above, the relationship between the experimental data and the data obtained was determined by means of the correlation coefficient. In addition, the optimization of the model was carried out using the method of least squares, having as an objective function the adjustment of the activation energy and the pre-exponential factor, which is achieved by means of the change of the slope and the cut-off point on the axis, respectively.

### 2.5. Validation of the Models

The validation of the models was carried out in the same way as a previous work carried out on the determination of the thermodynamic parameters of individual thermoplastics [19]. Thus, the optimization of the models has been accomplished using the least squares method, while the comparison and validation of the five proposed kinetic models is based on calculating the correlation coefficient between the theoretical and experimental data.

### 2.6. Thermodynamic Parameters

Catalytic pyrolysis from mixed plastic waste can be quantified by the energy changes that occur in the process using different conditions including heating rates, different catalysts and other recycled mixed plastic waste [18]. Moreover, the thermodynamic parameters provide information on the spontaneity of the catalytic pyrolysis process, in particular the Gibbs free energy (∆***G***) as a relevant factor that indicates the ease of pyrolysis of the plastic waste studied [20]. In this sense, the enthalpy change was determined using the following equation, which is based on the calculation of the activation energy related to the decomposition kinetics obtained:(14)$$∆H=E_{a}-RT$$

While the Gibbs free energy (∆***G***) and the entropy of the process (∆***S***) are calculated by
(15)$$∆G=E_{a}+RT_{m}*\ln\left(\frac{{K_{B}*T}_{m}}{h*A}\right)$$
(16)$$∆S=\frac{∆H-∆G}{T_{m}}$$
where ∆***G***, ***K_B_*** and h represent the Gibbs free energy, Boltzmann’s constant and Planck’s constant, respectively.

## 3. Results

### 3.1. FTIR Analysis

FTIR has been used for the chemical characterization of both plastic wastes, i.e., polypropylene (PP) and polystyrene (PS). The PP spectrum (Figure 1a) shows three groups of bands, which correspond to tension movements of the CH bonds at 2900 cm^−1^, DC tension movements at 1350–1470 cm^−1^ and a bending movement of -CH_3_ between 1200 and 1000 cm^−1^. The PS spectrum (Figure 1b) presents three groups of absorption bands, which appear at the multiple tension movements of the C-H bonds at 2700–3000 cm^−1^, C-C at 1400–1600 cm^−1^ of the aromatic ring and a bending movement of -CH_2_ and tensions of aromatic rings between 700–800 cm^−1^. In addition, both spectra obtained coincide with the results obtained in other works reported in the literature [21].

### 3.2. Thermogravimetric Analysis (TGA)

The activation energy values were obtained from TGA data processing at different heating rates using an established reaction model, which provides a general or macroscopic value, simplifying its determination since this process of degradation of mixed plastic waste presents a wide range of resulting products from a complexity of multiple reactions [10].

The thermocatalytic degradation profiles at three heating rates (5, 10 and 15 °K min^−1^) are shown in Figure 2a. It should be noted that catalytic activity does not affect the reaction mechanism, which occurs in a single step, similar to the sample without a catalyst (SC). However, the degradation temperature profiles obtained from plastic blends causes a leftward shift of the curve towards lower temperatures compared to the SC sample, resulting in a slight reduction of the maximum degradation temperature and thus a decrease in the activation energy [22]. In addition, when the catalyst is accompanied by slow heating with progressive and controlled heat transfer, it causes a faster loss of mass at lower temperatures and with higher speed [23].

In Figure 2b, the DTG curves illustrate that mixed plastic waste in the presence of FCC catalyst degrades at different temperature ranges using three heating rates as follows: (i) at 5 K min^−1^, it starts at 635 K and ends at approximately 716 K; (ii) at 10 °K min^−1^, the degradation range is between 643 and 731 K; and (iii) for a heating rate of 15 °K min^−1^, the degradation range is from 643 K to 733 K. These obtained values matched well with the degradation temperature range of pure plastics reported by other authors [24], where the decomposition temperature range for PP was between 688 and 813 K and for PS was between 645 and 725 K. However, this study (PP/PS, 25:75, wt%) presents a slightly lower degradation temperature range at the three heating rates, due to PP presenting a greater degradation at lower temperatures when PS is present, which is probably due to the acid catalytic action that helps some radicals originated in the degradation of PS interact with the PP present in the mixture. This results in the formation of compounds at slightly lower temperatures compared to the reaction without catalyst [25,26], which may be due to the transfer of a hydrogen atom from the less stable polymer to the other polymer at the time of contact with the catalyst present in the reaction [27].

It can also be observed that the application of the R2 degradation mechanism at different heating rates explains that the heat flow first decomposes the outer layers of the polymer; when penetrating to the inner layers, it decomposes the complex molecules into low molecular weight hydrocarbons, causing the degradation to move to a lower temperature zone—an effect that is favored by the presence of the catalyst (Figure 2a). Therefore, the presence of catalyst during the reaction has a positive effect on the yield of volatile components. For PP and PS, the thermal stability is affected by the type of carbocation it generates. The R2 degradation model assumes that the rate of the degradation reaction starts at the surface, and the rate is controlled by the progress of the interphase reaction favoring processes where the heating rate is slow [27]. Figure 2b shows that the decomposition form of the plastic mixture does not change upon the addition of catalyst observing a single reaction peak.

### 3.3. Kinetic Parameters

The determination of the overall activation energy of the degradation process is a chemical kinetics parameter that offers a comprehensive understanding of the catalytic effect in comparison to a non-catalytic process [19]. In all cases, each isoconversion model used at the three heating rates presented a high relationship between the experimental values and the corresponding data (very close to one), as shown in Figure 3. However, it a slight decrease in the data correlation is observed at the heating rate of 15 K min^−1^, which can be attributed to the intricate reactions taking place during the pyrolysis process, especially with the incorporation of a catalyst that generates more reactivity at higher temperatures through active points [7,18]. Nevertheless, it is evident that the integral models exhibit the most accurate fit for this catalytic process, demonstrating a high correlation with data greater than 0.90.

Table 1 shows the results obtained for the activation energy from 308 to 148 kJ/mol and the pre-exponential factor 7 × 10^−9^ to 1 × 10^−22^ with the corresponding equations of each model applying the contracting cylinder R2 mechanism, recommended by Dubdub et al. [14], which was found to be a suitable approach for this type of mixture.

Figure 4 shows the results of the activation energy for the five kinetic models studied. It can be noted that as the heating rate increases, the dispersion of data decreases significantly for each of the isoconversion methods. At 5 K min^−1^, the values range between 170 and 308 kJ mol^−1^; at 10 K min^−1^, the range is from 152 to 240 kJ mol^−1^; and at 15 K min^−1^, the values range from 148 to 173 kJ mol^−1^. Thus, it is evidence that the presence of the catalyst in the plastic waste mixture affects the decomposition rate but not the intramolecular reaction [27,28], which is corroborated by other works found in the literature without the use of a catalyst, with obtained activation energy values (*E_a_*) of 250 kJ mol^−1^ and 198 kJ mol^−1^ from PP/PS ratios (50:50) and (70:30), respectively. In addition, the rate of decomposition can be attributed to the transfer of radicals between different polymers through intermolecular reactions. The catalyst plays a crucial role in promoting and facilitating active sites during the reaction, leading to the cleavage of weak bonds and transfer of intermolecular chains [29].

However, the DTG curves from PP/PS plastic waste at different heating rates (Figure 2b–d) demonstrated a better fit in the data experimental at 15 K min^−1^ with respect to the obtained values from the isoconversional models studied with a similar peak intensity and a lower decomposition temperature compared to the other heating rates (5 and 10 K min^−1^), which is probably due to the generation of more heat flux, which facilitates shorter degradation times of all the components present in the mixture of plastic waste.

In this way, Martinez et al. [19] indicated that plastic mixtures at low heating rates (<20 K min^−1^) using only thermal pyrolysis show poor fits to the experimental data throughout the decomposition range, due to the different steps in which they decompose, especially since the steps show varying degrees of overlap at various heating rates [19,30]. However, by using a regenerated FCC catalyst, the DTG curves fit the experimental data well with a single kinetic triplet at heating rates of 15 K min^−1^ with certain inflection points. Thus, the methodology proposed in this study can be applied to any thermoplastic, i.e., PP, PE, PS, PET and PVC, or materials that do not have a similar structure under similar conditions.

### 3.4. Thermodynamic Parameters

Table 2 shows the data on enthalpy consumption, entropy changes and Gibbs free energy for the mixture of plastic waste at different heating rates with the data obtained from the Starink kinetic model, the model that presented the best fit.

According to Eyring’s theory, the enthalpy variation is directly proportional to the activation energy, which coincides with the values obtained in the degradation of plastic waste mixtures, with low energy consumption at 15 K min^−1^ and a value of Gibbs free energy (∆*G*) slightly lower compared to the other heating rates that provides greater ease of catalytically pyrolyzing mixtures of PP and PS plastics.

## 4. Conclusions

The present study involved the regeneration of an FCC catalyst through chemical and thermal treatment under suitable conditions for its subsequent use in catalytic pyrolysis from recycled mixtures of polypropylene and polystyrene (25:75, wt%). The kinetic behavior of this plastic waste mixture was evaluated using thermogravimetric analysis (TGA) at three heating rates (5, 10 and 15 K min^−1^) using 5 kinetic models that included FR, KAS, FWO, ST and MM. The DTG curves of the PP/PS plastic waste at 15 K min^−1^ show a better fit with those obtained from the kinetic models studied. However, the Starink integral isoconversional method shows the highest correlation coefficient (R2) obtaining activation energy values of 188, 215 and 148 kJ mol^−1^ at heating rates of 5,10 and 15 K min^−1^, respectively.

In addition, the thermodynamic parameters were calculated from the kinetic parameters of the reaction obtaining activation energy and pre-exponential factor values ranging from 308 to 148 kJ/mol (*E_a_*) and from 7 × 10^−9^ to 1 × 10^−22^ (*A*) for the 5 kinetic models. In this sense, the temperature and the proportion of catalyst used in this work have a significant impact on the catalytic pyrolysis which promotes and facilitates the cleavage of weak bonds and the transfer of intermolecular chains, generating a decrease in the activation energy compared to the process without catalyst under the same pyrolysis conditions.

## Figures and Tables

**Figure 1 polymers-15-02035-f001:**
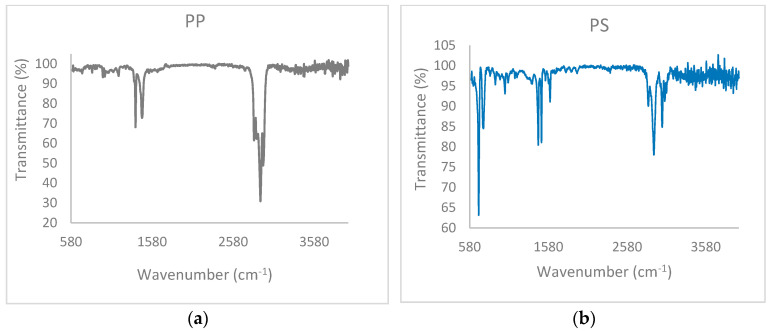
FTIR Spectra of raw (**a**) polypropylene and (**b**) polystyrene.

**Figure 2 polymers-15-02035-f002:**
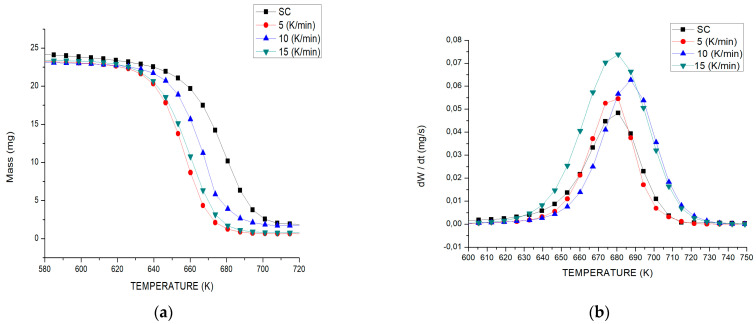
Catalytic degradation at three heating speeds (5, 10, 15 K min^−1^) of PP and PS mixture. (**a**) Profile for TGA and (**b**) DTG curves of different plastics.

**Figure 3 polymers-15-02035-f003:**
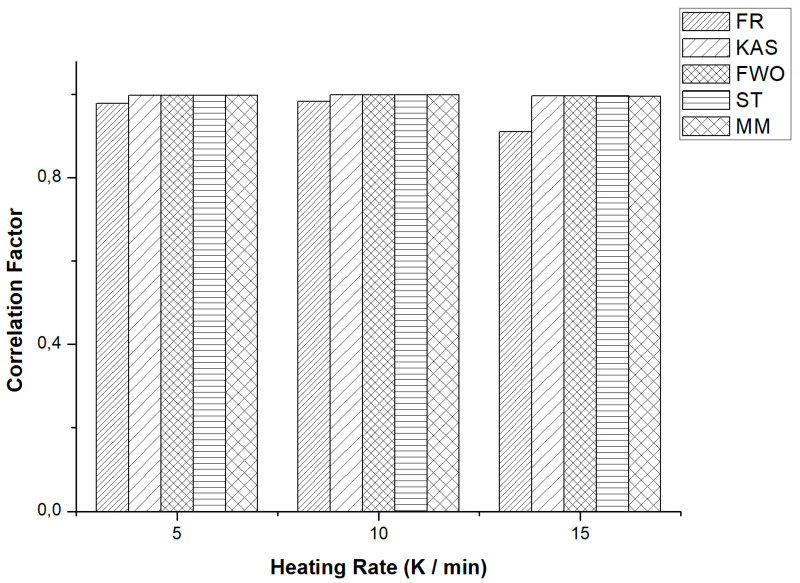
Comparison of the different isoconversional methods for different heating rates from a mixture of plastic waste PP and PS.

**Figure 4 polymers-15-02035-f004:**
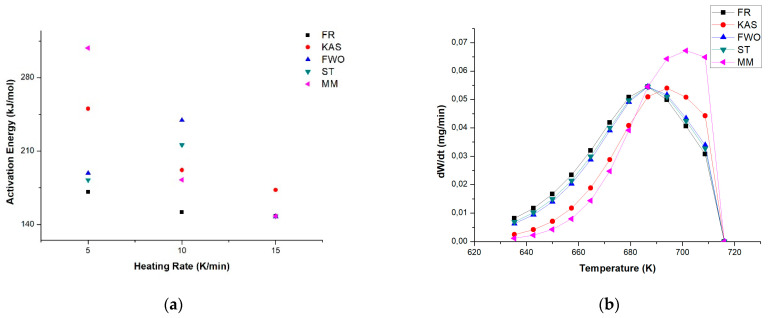
(**a**) Dispersion of data obtained in the activation energy obtained from the isoconversion methods. Catalytic degradation of PP and PS mixture at three heating speeds of (**b**) 5 K min^−1^, (**c**) 10 K min^−1^, and (**d**) 15 K min^−1^.

**Table 1 polymers-15-02035-t001:** Kinetic parameters obtained by isoconversional methods from mixture PP and PS.

Model	Heating Rate (K/min)	Model Equation	*E_a_* (KJ/mol)	*A* (s^−1^)
**KAS**	5	$\ln\frac{\left(1-{\left(1-\alpha \right)}^{\frac{1}{2}}\right)}{{T_{m}}^{2}}=32.893-\frac{30127}{T}$	250.474	4.841 × 10^17^
	10	$\ln\frac{\left(1-{\left(1-\alpha \right)}^{\frac{1}{2}}\right)}{{T_{m}}^{2}}=21.939-\frac{23081}{T}$	191.897	1.298 × 10^13^
	15	$\ln\frac{\left(1-{\left(1-\alpha \right)}^{\frac{1}{2}}\right)}{{T_{m}}^{2}}=18.455-\frac{20799}{T}$	172.921	5.380 × 10^11^
**FWO**	5	$\ln\left(1-{\left(1-\alpha \right)}^{\frac{1}{2}}\right)=37.192-\frac{23904}{T}$	188.916	1.076 × 10^13^
	10	$\ln\left(1-{\left(1-\alpha \right)}^{\frac{1}{2}}\right)=45.676-\frac{240}{T}$	239.596	4.102 × 10^16^
	15	$\ln\left(1-{\left(1-\alpha \right)}^{\frac{1}{2}}\right)=29.594-\frac{18692}{T}$	147.728	6.987 × 10^11^
**FR**	5	$\ln\left(\frac{d\alpha }{dt}\right)=35.857-\frac{20548}{T}$	170.836	4.533 × 10^11^
	10	$\ln\left(\frac{d\alpha }{dt}\right)=32.238-\frac{18542}{T}$	151.663	1.216 × 10^10^
	15	$\ln\left(\frac{d\alpha }{dt}\right)=30.662-\frac{17822}{T}$	148.172	7.448 × 10^9^
**ST**	5	$\ln\left(1-{\left(1-\alpha \right)}^{\frac{1}{2}}\right)=42.175-\frac{22880}{T}$	188.715	1.039 × 10^13^
	10	$\ln\left(1-{\left(1-\alpha \right)}^{\frac{1}{2}}\right)=45.909-\frac{26180}{T}$	215.937	8.604 × 10^14^
	15	$\ln\left(1-{\left(1-\alpha \right)}^{\frac{1}{2}}\right)=33.591-\frac{17963}{T}$	148.157	7.428 × 10^14^
**MM**	5	$\ln\frac{\left(1-{\left(1-\alpha \right)}^{\frac{1}{2}}\right)}{{T_{m}}^{2}}=42.890-\frac{37087}{T}$	308.343	1.309 × 10^22^
	10	$\ln\frac{\left(1-{\left(1-\alpha \right)}^{\frac{1}{2}}\right)}{{T_{m}}^{2}}=20.376-\frac{21962}{T}$	182.589	2.586 × 10^12^
	15	$\ln\frac{\left(1-{\left(1-\alpha \right)}^{\frac{1}{2}}\right)}{{T_{m}}^{2}}=14.303-\frac{17803}{T}$	148.012	7.245 × 10^9^

*E_a_*: Activation energy. *A*: Pre-exponential factor.

**Table 2 polymers-15-02035-t002:** Values obtained from the thermodynamic parameters determined with the Starink model at different heating rates of the PP and PS mixture.

*Β* (K min^−1^)	*E_a_* (kJ mol^−1^)	*A* (s^−1^)	∆*H* (kJ mol^−1^)	∆*G* (kJ mol^−1^)	∆*S* (kJ mol^−1^)
5	188.715	1.039 × 10^13^	1.830 × 10^2^	1.905 × 10^2^	−1.098 × 10^−2^
10	215.937	8.604 × 10^14^	2.102 × 10^2^	1.931 × 10^2^	−2.463 × 10^−2^
15	148.157	7.428 × 10^14^	1.424 × 10^2^	1.920 × 10^2^	−7.130 × 10^−2^

## Data Availability

The data presented in this study are available on request from the corresponding author.

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
