# Peer review of "Kinetic Analysis of Thermal Degradation of Recycled Polypropylene and Polystyrene Mixtures Using Regenerated Catalyst from Fluidized Catalytic Cracking Process (FCC)"

_polymers, 2023, doi:10.3390/polym15092035_

Round 1

Reviewer 1 Report

Thank you for the manuscript. I believe that the idea of this study has potential to be published, although I would recommend some changes in the manuscript. Suggestions to improve the quality of manuscript are as follows:

·       First letter of all the keywords should be needs to be written in uppercase.

·       Add 3-4 highlights of the manuscript.

·       Highlights should summarize the research methodology and findings.

·       Restructure the second sentence of introduction, and correct all other similar mistakes.

·       Focus on the novelty of the work while presenting highlights.

·       Keywords should be arranged in chronological order.

·       The objectives of the study should be clearly stated in the initial section. This will help the reader focus on the priorities.

·       Add the list of abbreviations.

·       The introduction section in unnecessarily lengthy and needs to be precise. The

information flow is not clear.

·       What research gaps does the current paper fill?

·       There is no validation and comparison of proposed methodology.

·       How could the methodology be used in other similar materials?

·       The paper is quite interesting but the sections and subsections make it difficult to read and follow. I would suggest, to re-structure it to make it more readable.

·       There are many places that requires citation.

·       Avoid repetition of sentences.

·       The literature review ignores many important and recent works.

·       Improve the scientific discussion on the long-term retention phase and compare the results with other studies.

·       The research gaps have not been identified and clarified.

·       Conclusions are difficult to read and do not summarize the main findings of the paper.

·       Check Reference formatting. Check POLYMERS guidelines.

·       Please ensure that every reference cited in the text and the figure(s) is also in the reference list (and vice versa).

·       Please go through the following research articles for better understanding of the concerned topic:

            https://doi.org/10.1016/j.biortech.2022.127770

          https://doi.org/10.1016/j.ijhydene.2021.12.177

             If it is appropriate then only, you can site these

Author Response

SUGGESTIONS REVIEWER 1:

We would like to thank Reviewer 1 for its kind questions and comments to improve the paper.

Please find next the reply to its requirements.

  1. First letter of all the keywords should be needs to be written in uppercase.

Reply: It has been modificated.

Keywords: Plastic waster; Regenerated FCC catalyst, Kinetic analysis, Thermocatalytic degradation, Isoconversional models.

  1. Add 3-4 highlights of the manuscript.

Reply: In the text has been added the main highlights to focus on the novelty of the work, for example the ideas following:

a.- In any case, all these studies have been carried out by means of thermal pyrolysis, which shows even more limited studies of the kinetic behavior using catalysts, being a field still to be explored due to its great advantages as mentioned above.

b.- In the sense, to understand kinetic behavior of plastic waste mixtures when degraded is very important, which is commonly evaluated by thermogravimetric analysis (TGA) under non-isothermal (dynamic) conditions, using different heating rates to determine their general or macroscopic kinetics of the thermocatalytic process.

c.- Figure 4 shows the results of the activation energy for five kinetic models studied. It can be noted that as the heating rate increases, the dispersion of data decreases sig-nificantly for each of the isoconversion methods. At 5 K min-1, the values range be-tween 170 and 308 kJ mol-1, at 10 K min-1 the range is from 152 to 240 kJ mol-1, and at 15 K min-1, the values range from 148 to 173 kJ mol-1.

e.- , each isoconversion model used at three heating rates presented a high relationship between the experimental values and the corresponding data (very close to one).

  1. Highlights should summarize the research methodology and findings.

Reply: In the text has been added the main highlights to focus on the novelty of the work, for example the ideas following:

a.- In any case, all these studies have been carried out by means of thermal pyrolysis, which shows even more limited studies of the kinetic behavior using catalysts, being a field still to be explored due to its great advantages as mentioned above.

b.- In the sense, to understand kinetic behavior of plastic waste mixtures when degraded is very important, which is commonly evaluated by thermogravimetric analysis (TGA) under non-isothermal (dynamic) conditions, using different heating rates to determine their general or macroscopic kinetics of the thermocatalytic process.

c.- Figure 4 shows the results of the activation energy for five kinetic models studied. It can be noted that as the heating rate increases, the dispersion of data decreases sig-nificantly for each of the isoconversion methods. At 5 K min-1, the values range be-tween 170 and 308 kJ mol-1, at 10 K min-1 the range is from 152 to 240 kJ mol-1, and at 15 K min-1, the values range from 148 to 173 kJ mol-1.

e.- , each isoconversion model used at three heating rates presented a high relationship between the experimental values and the corresponding data (very close to one).

  1. Restructure the second sentence of introduction, and correct all other similar mistakes.

Reply: It has been modificated

Borelle et al. (2) indicates that by the year 2050 there will be a generation of about 25,000 million metric tons of plastic waste, of which 36.4% will end up in landfills or in the en-vironment; a similar fraction will be incinerated, and only 27.2% will be recycled.

  1. Focus on the novelty of the work while presenting highlights.

Reply: It has been included

….the present work represents the first kinetic study in report information on the behavior of the degradation of plastic waste mixtures under the catalytic pyrolysis conditions used to evaluate the relationship between the experimental values of TGA analysis with the corresponding data obtained from five isoconversion models studied, which includes two additional kinetic models compared to other works reported in the literature….

  1. Keywords should be arranged in chronological order.

Reply: It has been modificated.

Keywords: Plastic waster; Regenerated FCC catalyst, Kinetic analysis, Thermocatalytic degradation, Isoconversional models.

  1. The objectives of the study should be clearly stated in the initial section. This will help the reader focus on the priorities.

Reply: It has been modificated.

In this context, the main objective of this study is to analyze the kinetic degradation behavior under catalytic pyrolysis conditions from a mixture of recycled PP and PS (25:75, wt.%) using a regenerated fluidized catalytic cracking (FCC) catalyst from a petrochemical industry by thermo-gravimetric analysis (TGA). Moreover, the thermodynamic parame-ters were calculated from the kinetic analysis using five isoconversional models at three heating rates (5,10 and 15 K min-1) commonly used at industrial level, whose results were also compared without the use of catalyst. Thus, the design parameters, implementation and operating conditions of catalytic pyrolysis can be evaluated by analyzing its chemical kinetics whose accuracy and precision depend on the reliability in the calculation of its activation energy and pre-exponential factor [7]. Further, the present work represents the first kinetic study in report information on the behavior of the degradation of plastic waste mixtures under the catalytic pyrolysis conditions used to evaluate the relationship be-tween the experimental values of TGA analysis with the corresponding data obtained from five isoconversion models studied.

  1. Add the list of abbreviations.

Reply: In the text, the following abbreviations are described after the name. Similarly, each equation has its description. For instance:

Aboulkas et al. [7] determined the activation energy for different plastic residues considered the most abundant in domestic garbage, that includes high-density polyethylene (HDPE), reporting values between 238 and 247 kJ mol-1, polyethylene low density (LDPE) from 215 to 221 kJ mol-1 and polypropylene (PP) between 179 and 188 kJ mol-1 [6,11,12]. Moreover, polystyrene (PS)……….

Single-use plastic waste that included polypropylene (PP) and polystyrene (PS) were recycled through completely random sampling of a garbage dump located in Riobamba, Ecuador. These samples were classified and the PP was separated from the PS. Then, these were washed and crushed to a particle size of 1 mm. The catalyst was collected from a fluidized catalytic cracking (FCC)…..

Fourier Transform Infrared Spectroscopy (FTIR)……

  1. The introduction section in unnecessarily lengthy and needs to be precise. The information flow is not clear.

Reply: It has been modificated.

Currently, the solid waste generated at the urban and industrial level has intensified in recent years, being plastics one of the wastes of greatest concern worldwide due to sev-eral factors that include the high consumption of resources for its production, its high daily use and its long degradation time due to its polymeric structure (1). Borelle et al. (2) indicates that by the year 2050 there will be a generation of about 25,000 million metric tons of plastic waste, of which 36.4% will end up in landfills or in the environment; a sim-ilar fraction will be incinerated, and only 27.2% will be recycled. Thus, it is essential to search for new sustainable technological alternatives such as pyrolysis that contribute to environmentally adequate management of plastic waste [3–5]. Pyrolysis is a thermo-chemical process that involves the degradation of plastics at high temperatures in the ab-sence of oxygen to obtain valuable products that include liquid and gaseous hydrocar-bons, which can occur with or without the presence of catalyst [XX]. The main advantage of catalytic pyrolysis is to accelerate the reaction rate by reducing the activation energy of the process, which favors this process by increasing its efficiency [6]. In addition, this pro-cess can be carried out on both individual plastics and plastic mixtures, however, its ki-netic behavior may vary depending on their composition, which causes additional com-plexity due to the possible interaction between the different types of plastics that make up the mixture and to the formation of additional degradation products [XX]. In this way, Aboulkas et al. (7) determined the activation energy for different plastic residues consid-ered the most abundant in domestic garbage, that includes high-density polyethylene (HDPE), reporting values between 238 and 247 kJ mol-1, polyethylene low density (LDPE) from 215 to 221 kJ mol-1 and polypropylene (PP) between 179 and 188 kJ mol-1 [6,11,12]. Moreover, polystyrene (PS) is one of the most difficult plastics to recycle due to its lower thermal stability with an activation energy value of around 140 kJ mol-1 (13,14). However, studies of the behavior of plastic mixtures are more limited, being the PE and PP mixtures the most studied at different rates and different compositions, where minimum amounts of PP generate a decrease in the activation energy and in the maximum degradation tem-perature both for HDPE and LDPE, which is due to the synergy effect of the mixture influ-enced mainly by the lower thermal stability of PP in these composites [15,16]. In addition, Dubdub et al. [17] shows the behavior of different plastics and their mixtures where the data obtained is adjusted to a single stage of thermal degradation and the pyrolysis of pure plastic waste occurs in the order PS < PP < LDPE < HDP, while specific mixtures of PP and PS (50/50,wt%) present lower degradation with activation energy values between 144 and 188 kJ mol-1 [XX]. In any case, all these studies have been carried out by means of thermal pyrolysis, which shows even more limited studies of the kinetic behavior using catalysts, being a field still to be explored due to its great advantages as mentioned above.

In the sense, to understand kinetic behavior of plastic waste mixtures when degraded is very important, which is commonly evaluated by thermogravimetric analysis (TGA) under non-isothermal (dynamic) conditions, using different heating rates to determine their general or macroscopic kinetics of the thermocatalytic process [8]. According to In-ternational Confederation of Thermal Analysis Calorimetry (ICTAC) [9,10], the kinetic tri-plet of the pyrolytic process, i.e., activation energy, reaction order, and pre-exponential factor can be calculated from recommended isoconversional models with the data ob-tained from the TGA. Different isoconversion models that included Friedman, Kissin-ger-Akahira-Sunose (KAS) and Flynn-Wall-Ozawa (FWO) have been used to evaluate the kinetic behavior of plastic waste [XX], and the method of Criado for the calculation of the reaction mechanism [XX]. Thus, the study of kinetics plays a key role in the pyrolysis pro-cess on an industrial scale, even when catalytic conditions are handled, where different techniques and catalysts can be used, including those recovered from conventional pro-cesses, thus to improve the understanding of the thermocatalytic degradation processes and to develop effective recycling and plastic waste management strategies is essential.

In this context, the main objective of this study is to analyze the kinetic degradation behavior under catalytic pyrolysis conditions from a mixture of recycled PP and PS (25:75, wt.%) using a regenerated fluidized catalytic cracking (FCC) catalyst from a petrochemical industry by thermo-gravimetric analysis (TGA). Moreover, the thermodynamic parame-ters were calculated from the kinetic analysis using five isoconversional models at three heating rates (5,10 and 15 K min-1) commonly used at industrial level, whose results were also compared without the use of catalyst. Thus, the design parameters, implementation and operating conditions of catalytic pyrolysis can be evaluated by analyzing its chemical kinetics whose accuracy and precision depend on the reliability in the calculation of its activation energy and pre-exponential factor [7]. Further, the present work represents the first kinetic study in report information on the behavior of the degradation of plastic waste mixtures under the catalytic pyrolysis conditions used to evaluate the relationship be-tween the experimental values of TGA analysis with the corresponding data obtained from five isoconversion models studied.

  1. What research gaps does the current paper fill?

Reply: It has been included.

Further, the present work represents the first kinetic study in report information on the behavior of the degradation of plastic waste mixtures under the catalytic pyrolysis conditions used to evaluate the relationship between the experimental values of TGA analysis with the corresponding data obtained from five isoconversion models studied.

  1. There is no validation and comparison of proposed methodology.

Reply: this paragraph was included, the method of data validation is included here.

The validation of the models was carried out in the same way as a previous work carried out on the determination of the thermodynamic parameters of individual thermoplastics . Thus, the optimization of the models has been accomplished using the least squares method, while the comparison and validation of the five proposed kinetic models is based on calculating the correlation coefficient between the theoretical and experimental data.

  1. How could the methodology be used in other similar materials?

Chemical kinetics is the starting point for the design of any reactor or catalytic process, obtaining data on the degradation of this material. For this reason, its study is fundamental both in thermal and catalytic processes of one or several plastics. In this sense, the methodology proposed in this study can be applied to any thermoplastic, i.e., PP, PE, PS, PET and PVC or materials that do not have a similar structure under similar conditions.

  1. The paper is quite interesting but the sections and subsections make it difficult to read and follow. I would suggest, to re-structure it to make it more readable.

Reply: the entire paper has been reviewed

  1. There are many places that requires citation.

Reply: This observation has been included

  1. Avoid repetition of sentences.

Reply: It has been modificated.

  1. The literature review ignores many important and recent works.

Reply: References from recent years have been included in the manuscript.

  1. Improve the scientific discussion on the long-term retention phase and compare the results with other studies.

Reply: The writing of the article has been improved for a better understanding of the reader. In addition, the results were compared with other studies.

The determination of the overall activation energy of the degradation process is a chemical kinetics parameter that offers a comprehensive understanding of the catalytic effect in comparison to a non-catalytic process [19]. In all cases, each isoconversion model used at three heating rates presented a high relationship between the experimental values and the corresponding data (very close to one), as shown in Figure 3. However, it is ob-served a slight decrease in the data correlation at heating rate of 15 K min-1, which can be attributed to the intricate reactions taking place during the pyrolysis process, especially with the incorporation of a catalyst that generates more reactivity at higher temperatures through active points [28,29]. Nevertheless, it is evident that the integral models exhibit the most accurate fit for this catalytic process, demonstrating a high correlation with data greater than 0.90……

Figure 4 shows the results of the activation energy for five kinetic models studied. It can be noted that as the heating rate increases, the dispersion of data decreases signifi-cantly for each of the isoconversion methods. At 5 K min-1, the values range between 170 and 308 kJ mol-1, at 10 K min-1 the range is from 152 to 240 kJ mol-1, and at 15 K min-1, the values range from 148 to 173 kJ mol-1. Thus, it is evidence that the presence of the catalyst in the plastic waste mixture affects the decomposition rate, but not the intramolecular re-action [27,30], which was corroborated by other works found in literature without the use of catalyst, with obtained activation energy values (Ea) of 250 kJ mol-1 and 198 kJ mol-1 from PP/PS ratios (50:50) and (70:30), respectively……….

  1. The research gaps have not been identified and clarified.

Reply: It has been modificated.

  1. Conclusions are difficult to read and do not summarize the main findings of the paper.

Reply: It has been modificated.

The present study involved the regeneration of an FCC catalyst through chemical and thermal treatment under suitable conditions for its subsequent use in catalytic pyrolisis from recycled mixtures of polypropylene and polystyrene (25:75, wt%). The kinetic behav-ior of this plastic waste mixture was evaluated using thermogravimetric analysis (TGA) at three heating rates (5, 10 and 15 K min-1) using five kinetic models that included: FR, KAS, FWO, ST and MM. The DTG curves of the PP/PS plastic waste at 15 K min-1 show a better fit with those obtained from the kinetic models studied. However, the STARINK integral isoconversional method shows the highest correlation coefficient (R2) obtaining activation energy values of 188, 215, and 148 kJ mol-1 at heating rates of 5,10, and 15 K min-1, respec-tively.

In addition, the thermodynamic parameters were calculated from the kinetic param-eters of the reaction obtaining of activation energy and pre-exponential factor values ranged from 308 to 148 kJ/mol (Ea) and from 7x10-9 to 1x10-22 (A) for the five model kinetic. In this sense, the temperature and the proportion of catalyst used in this work have a sig-nificant impact on the catalytic pyrolysis which promotes and facilitates the cleavage of weak bonds and the transfer of intermolecular chains, generating a decrease in the activa-tion energy compared to the process without catalyst under the same pyrolysis conditions

  1. Check Reference formatting. Check POLYMERS guidelines.

Reply: the entire paper has been reviewed

  1. Please ensure that every reference cited in the text and the figure(s) is also in the reference list (and vice versa).

Reply: the entire paper has been reviewed

  1. Please go through the following research articles for better understanding of the concerned topic:

https://doi.org/10.1016/j.biortech.2022.127770

https://doi.org/10.1016/j.ijhydene.2021.12.177

  1. If it is appropriate then only, you can site these

Reply: The references kindly suggested by the reviewer have been analysed, the manuscript has been modified following the guidelines of other manuscripts. However, these references were not added because the area of knowledge is different.

Reviewer 2 Report

The paper is well-written and organized. Kinetic analysis of thermal degradation of recycled PP and PS mixtures using regenerated catalyst from FCC process was studied. Different kinetic methods using TGA for pyrolysis of plastics were compared and the best STARINK model was selected based on its highest R2 value. However, proposed degradation mechanisms and their detailed interpretation based on the kinetic studies are not given. Combined kinetic analysis using different thermal schedules should also be outlined and discussed.

Author Response

SUGGESTIONS REVIEWER 2:

We would like to thank Reviewer 2 for its kind questions and comments to improve the paper.

Please find next the reply to its requirements.

The paper is well-written and organized. Kinetic analysis of thermal degradation of recycled PP and PS mixtures using regenerated catalyst from FCC process was studied. Different kinetic methods using TGA for pyrolysis of plastics were compared and the best STARINK model was selected based on its highest R2 value. However, proposed degradation mechanisms and their detailed interpretation based on the kinetic studies are not given. Combined kinetic analysis using different thermal schedules should also be outlined and discussed.

Reply: The discussion and the entire manuscript were reviewed and revised. In addition, this paragraph was included.

It can also be observed that the application of the R2 degradation mechanism at different heating rates explains that the heat flow first decomposes the outer layers of the polymer, when penetrating to the inner layers it decomposes the complex molecules into low molecular weight hydrocarbons, causing the degradation to move to a lower temperature zone, an effect that is favored by the presence of the catalyst (Fig. 2a). Therefore, the presence of catalyst during the reaction has a positive effect on the yield of volatile components. For PP and PS the thermal stability is affected by the type of carbocation it generates, the R2 degradation model assumes that the rate of the degradation reaction starts at the surface and the rate is controlled by the progress of the interphase reaction favoring processes where the heating rate is slow (27). Fig. 2b shows that the decomposition form of the plastic mixture does not change upon addition of catalyst observing a single reaction peak.

Round 2

Reviewer 2 Report

The manuscript has been improved sufficiently.